# Changes in Ammonia-Oxidizing Archaea and Bacterial Communities and Soil Nitrogen Dynamics in Response to Long-Term Nitrogen Fertilization

**DOI:** 10.3390/ijerph19052732

**Published:** 2022-02-26

**Authors:** Aixia Xu, Lingling Li, Junhong Xie, Subramaniam Gopalakrishnan, Renzhi Zhang, Zhuzhu Luo, Liqun Cai, Chang Liu, Linlin Wang, Sumera Anwar, Yuji Jiang

**Affiliations:** 1State Key Laboratory of Aridland Crop Science, Gansu Agricultural University, Lanzhou 730070, China; xuax@gsau.edu.cn (A.X.); xiejh@gsau.edu.cn (J.X.); zhangrz@gsau.edu.cn (R.Z.); luozz@gsau.edu.cn (Z.L.); cailq@gsau.edu.cn (L.C.); liuc@gsau.edu.cn (C.L.); wangll@gsau.edu.cn (L.W.); 2College of Agronomy, Gansu Agricultural University, Lanzhou 730070, China; 3International Crops Research Institute for the Semi-Arid Tropics (ICRISAT), Patancheru, Hyderabad 502324, Telangana, India; s.gopalakrishnan@cgiar.org; 4College of Resource and Environment, Gansu Agricultural University, Lanzhou 730070, China; 5Institute of Molecular Biology and Biotechnology, The University of Lahore, Lahore 54000, Pakistan; anwer_sumera@yahoo.com; 6State Key Laboratory of Soil and Sustainable Agriculture, Institute of Soil Science, Chinese Academy of Sciences, Nanjing 210008, China

**Keywords:** ammonia-oxidizing bacteria (AOB), ammonia-oxidizing archaea (AOA), nitrogen use efficiency (NUE), soil properties

## Abstract

Ammonia oxidizing archaea (AOA) and bacteria (AOB) mediate a crucial step in nitrogen (N) metabolism. The effect of N fertilizer rates on AOA and AOB communities is less studied in the wheat-fallow system from semi-arid areas. Based on a 17-year wheat field experiment, we explored the effect of five N fertilizer rates (0, 52.5, 105, 157.5, and 210 kg ha^−1^ yr^−1^) on the AOA and AOB community composition. This study showed that the grain yield of wheat reached the maximum at 105 kg N ha^−1^ (49% higher than control), and no further significant increase was observed at higher N rates. With the increase of N, AOA abundance decreased in a regular trend from 4.88 × 10^7^ to 1.05 × 10^7^ copies g^−1^ dry soil, while AOB abundance increased from 3.63 × 10^7^ up to a maximum of 8.24 × 10^7^ copies g^−1^ dry soil with the N105 treatment (105 kg N ha^−1^ yr^−1^). Application rates of N fertilizer had a more significant impact on the AOB diversity than on AOA diversity, and the highest AOB diversity was found under the N105 treatment in this weak alkaline soil. The predominant phyla of AOA and AOB were *Thaumarchaeota* and *Proteobacteria*, respectively, and higher N treatment (N210) resulted in a significant decrease in the relative abundance of genus *Nitrosospira*. In addition, AOA and AOB communities were significantly associated with grain yield of wheat, soil potential nitrification activity (PNA), and some soil physicochemical parameters such as pH, NH_4_-N, and NO_3_-N. Among them, soil moisture was the most influential edaphic factor for structuring the AOA community and NH_4_-N for the AOB community. Overall, 105 kg N ha^−1^ yr^−1^ was optimum for the AOB community and wheat yield in the semi-arid area.

## 1. Introduction

Nitrification is the most crucial step in the biogeochemical nitrogen (N) cycle in which NH_3_ is oxidized to NO_2_^−^ and then to NO_3_^−^ [1,2,3]. The nitrification process regulates the absorption and availability of N to crops in the soil ecosystem [4,5] and may also cause the loss of N because leaching of NO_3_^−^ is much easier than NH_3_ [6]. Ammonia oxidizing archaea (AOA) and bacteria (AOB) mediate a very critical step in nitrogen (N) metabolism [7,8]. Both AOA and AOB contain the gene encoding the ammonia monooxygenase *amoA*, which has the catalytic ability to oxidize ammonia to hydroxylamine, which can be used as a marker for evaluating AOA and AOB abundances [9]. However, AOA and AOB communities’ response to environmental drivers is still not completely understood [10,11].

Nitrogen is the main element and a limiting nutrient for sustainable agricultural production [12,13]. The application of chemical N fertilizers, such as urea or ammonium compounds, is the main source of N in the agricultural system worldwide [14]. Long-term N application not only drives the loss of biodiversity in the agricultural ecosystem [15,16,17] but also usually regulates the continuous increase of soil potential nitrification activity (PNA) [18]. The change of PNA is considered to be closely related to soil pH and the community of ammonia oxidizers [19]. Fertilization affects the diversity and structure of AOB and AOA communities, however, there are still few studies on long-term inorganic N fertilization affecting ammonia oxidation in agricultural soils [20]. More than one-third of the world’s production and consumption of N fertilizer is in China [21]. The extensive input of ammonium fertilizers (such as urea) has been reported to selectively stimulate the AOA or AOB abundance in the agricultural field [19], and in turn, microorganism-mediated nitrification ultimately leads to a reduction of nitrogen use efficiency (NUE) [22,23]. Moreover, excessive N fertilizer application not only increases the risk of nitrate (NO_3_-N) leaching and runoff loss [24,25], and loss of N_2_O or N_2_ due to denitrification and ammonia oxidation [26], but also leads to severe pollution [27] and greenhouse effect [28,29], which have been major environmental problems in some regions of Europe, the United States, and China [30,31,32]. Therefore, most countries in the EU have taken measures to limit the rate of N application [33,34]. Thus, there is an urgent need to understand the response of ammonia oxidizers to N fertilizer applications and to predict the effectiveness and safety of N fertilizer applications [26,35]. 

AOA and AOB coexist in various soil types, coping with environmental disturbances and resource utilization in different ways [11,36]. Some studies have confirmed that the AOA community dominates agricultural soils [37,38,39] and is a primary driving force of nitrification [40,41]. However, some studies have also shown that although AOA has an absolute advantage in quantity, AOB has a dominant position in function [42,43]. In diverse soil ecosystems, the ratio of soil AOA to AOB in semiarid ecosystems is from 17 to 1600 [6]. Studies have shown that soil N availability and pH are key factors in constructing AOA and AOB communities, AOA prefers less ammonia and acidic conditions, while AOB prefers nitrogen-rich alkaline conditions [19,42]. In addition, temperature [44] and O_2_ [45] are also effective factors affecting AOA and AOB communities. The oxidation of ammonia is traditionally thought to be carried out by AOB, so N-based fertilization may be a primary driver for AOB [46]. However, recent studies have shown that AOA are also ammonia-oxidizing [47].

Long-term application of chemical N fertilizer will cause changes in soil physicochemical parameters [48]. However, there is still a lack of evidence on how long-term different N gradients affect the soil properties, and thus the activity, composition, and AOA and AOB abundance in semi-arid areas. This study hypothesized that N fertilization affects soil parameters and AOA and AOB abundance, thereby influencing crop yield and N utilization efficiency. Therefore, this study selected a 17-year wheat field experiment with different rates of N application, assessed the abundance of AOA and AOB communities by real-time quantitative PCR (qPCR), and assessed the community composition and structure of AOA and AOB by Illumina sequencing. The purpose was to explore: (i) the effect of N fertilization on the abundance of soil ammonia-oxidizing archaea (AOA) and ammonia-oxidizing bacteria (AOB) communities; (ii) the effect of N fertilization on the diversity and community composition of AOA and AOB communities; and (iii) the influence relationship and mechanism among N fertilization, soil physicochemical parameters, PNA, wheat yield, and NUE. Our research will deepen understanding soil factors in driving AOA and AOB diversity, N utilization efficiency, and N management in semiarid soil agricultural production.

## 2. Materials and Methods

### 2.1. Site Description and Experimental Design

The field site was located at the Dingxi Experimental Station of Gansu Agricultural University, Dingxi, Gansu Province, China (35°28′ N, 104°44′ E, elevation 1971 m a.s.l.). The soil for the field trials was Calcaric Cambisoll [49], locally called Huangmian [50]. 

Nitrogen fertilizer (urea) was applied every year from 2003 to 2019 before spring wheat sowing. The five N rates, i.e., N0 (non-N-fertilized control), N52.5 (52.5 kg N ha^−1^), N105 (105.0 kg N ha^−1^), N157.5 (157.5 kg N ha^−1^), and N210 (210.0 kg N ha^−1^), and three replications of each N were arranged in a randomized block design. The area of each test plot was 10 m × 3 m. The calcium superphosphate (105 kg P_2_O_5_ ha^−1^) was applied to all plots. Before sowing, the fertilizers were spread evenly over the entire cultivated area and incorporated into the 0–20 cm soil layer with a rotary tiller. Spring wheat (Dingxi No. 38) was sown in mid-March at a row spacings of 20 cm and a sowing density of 187.5 kg ha^−1^. The basic physicochemical parameters of soil from 0–100 cm depth, recorded in 2003 were: 0.78 g kg^−1^ total N, 4.92 mg kg^−1^ NH_4_-N, 27.12 mg kg^−1^ NO_3_-N, 8.33 pH, 5.98 mg kg^−1^ available phosphorus, 1.80 g kg^−1^ total phosphorus, 18.37 g kg^−1^ total potassium, 201.43 mg kg^−1^ available potassium, 1.20 g cm^−3^ bulk density, and 12.27 g kg^−1^ organic matter. The field site’s average daily minimum (January) and maximum (July) temperatures from 2003 to 2019 were −22 °C and 38 °C, respectively, with 7.4 °C mean annual temperature, 390.7 mm yr^−1^ mean annual precipitation, 1531 mm mean annual evaporation, and 5930 MJ m^−2^ mean annual radiation.

### 2.2. Sampling and Physicochemical Analyses

During the flowering stage of wheat in 2019, soil samples were collected from each plot using an auger. Each sample was composed of 5 random cores of 0–20 cm soil layers, mixed and separated into two sub-samples. One sub-sample was used to determine physicochemical parameters analyses, and the second subsample was used for molecular analyses.

Soil total nitrogen (TN) was analyzed using the method of Semimicro–Kjeldahl digestion analyses [51], while NH_4_-N and NO_3_-N were analyzed using the method of spectrophotometry [52]. The soil pH was analyzed using a glass combination electrode in a suspension of soil and water (ratio 1:2.5) [53]. The available phosphorus and total phosphorus in the soil were measured using the molybdenum antimony colorimetric method [51] and soil moisture (SM) was measured using the drying method [54]. Soil potential nitrification activity (PNA) was analyzed using the method of chlorate inhibition [55].

### 2.3. DNA Extraction and qPCR

The total DNA was extracted using an EZNA^®^ soil DNA kit (Omega, cat: M5635-02, CA, USA), and the quality of the extraction was checked by 1.2% agarose gel electrophoresis and then measured with a spectrophotometer (NanoDrop 2000, Thermo, New York, NY, USA) to quantify. Abundance of AOA and AOB were determined by the SYBR Green dye method real-time PCR experiment (mRNA) (absolute quantification PCR, AQ-PCR). All qPCR reactions were performed using a Q5^®^ High-Fidelity DNA Polymerase kit (M0491, Gene Biotechnology International Trade Co., Ltd., Shanghai, China) with TIB8600 PCR System. The primers for AOA gene abundance were Arch-amoA26F (5′-GACTACATMTTCTAYACWGAYTGGGC-3′) and Arch-amoA417R (5′-GGKGTCATRTATGGWGGYAAYGTTGG-3′) [56], while primers for AOB gene abundance were F (5′-GGGGTTTCTACTGGTGGT-3′) and R (5′-CCCCTCKGSAAAGCCTTCTTC-3′) [57]. The standard curve with known copy number was used as the standard curve, and the initial copy number of the unknown sample DNA was obtained by measuring the threshold cycle (Ct) value of the unknown sample and combining with the standard curve. The gene copy numbers of *amoA* AOA and AOB were recorded as copies in one g soil. The amplification efficiencies of AOA and AOB were 87.8% and 82.5%, respectively, and R^2^ values were 0.9993 and 0.9995, respectively. The copy number (X_0_) was calculated as follows:Ct = −K logX_0_ + b(1)

### 2.4. Illumina Miseq Sequencing and Bioinformatic Analysis

The sequences of the PCR products were obtained using the Illumina MiSeq platform and the primers described above. The PCR amplified product was quantified with the Quant-iT PicoGreen dsDNA Assay Kit (Shanghai Yanhui Biotech. Co., Ltd., Shanghai, China) and a microplate reader (FLx800 BioTek). The qualified computer sequencing libraries were mixed in corresponding proportions according to the required sequencing amount and were denatured into single strands with NaOH for computer sequencing. The MiSeq Reagent Kit V3 (600 cycles, Shenzhen Hisian Biotechnology Co., Ltd., Shenzhen, China) reagent and MiSeq sequencer were used for the double-end measurement. The optimal sequencing insert range was kept as 200 to 450 bp to limit the read length of MiSeq sequencing while ensuring sequencing quality. The sequence was denoised using the QIIME2 dada2 analysis, and the remaining were clustered into operational classification units (OTU) using the Vsearch software analysis and used for subsequent analysis. 

After quality filtering using QIIME2, we obtained 611,587 *amoA* AOA and 321,422 *amoA* AOB sequences among 15 soil samples. The number of AOA sequences in each sample ranged from 29,767 to 69,464, and the AOB ranged from 15,688 to 25,333. The sequences of AOA and AOB were rarefied to 28,278 and 14,903 per sample, respectively.

### 2.5. Statistical Analysis

One-way ANOVA with Duncan testing was used to determine the significance of N rates on the wheat grain yield (GY), NUE, soil physicochemical parameters, the abundance of AOA, AOB, and alpha diversity. The correlations among soil physicochemical parameters, PNA, AOA, and AOB abundance were evaluated by Pearson’s correlation coefficient analysis. The differences between the community structure of AOA and AOB under N fertilizer treatments were calculated using principal coordinate analysis (PCoA). Canoco 5.0 was used to analyze the relationships between soil physicochemical traits and AOA and AOB abundance. The relationship between soil environmental parameters and the structure of *amoA* AOA and *amoA* AOB was coordinated by Redundancy Analysis (RDA). Forward selection was conducted to test the significance of soil physicochemical parameters on the AOA and AOB community structure. The network was constructed from all soil samples taken together.

## 3. Results

### 3.1. Soil Physicochemical Parameters, PNA, Yield of Wheats, and NUE

The soil physicochemical properties were significantly altered after 17 years of different N fertilization rates (Table 1). The soil was weakly alkaline, and the pH was decreased from 8.99 to 8.67 (decrease by 2.04–3.58%) with increasing fertilizer rates. A significant difference in soil pH was recorded between N0 and N, and soil pH at N52.5 was significantly higher than that at N157.5 and N210. Soil NH_4_-N and NO_3_-N were increased by 0.91–55.20% and 25.62–175.85% by increasing N rates, respectively. Available phosphorus (AP) was significantly enhanced (23.29%) by the application of 105 kg N ha^−1^ compared with no-N control. No significant difference in total phosphorus, total N, and moisture contents was observed with N addition. 

The rates of N fertilizer application significantly affected the potential nitrification activity (PNA) (Table 1). The PNA ranged from 1.22 to 2.32 mg NO_3_-N g^−1^ h^−1^, and the highest PNA was obtained under the N105 treatment. In addition, under the N52.5, N105, N157.5, and N210 treatments, PNA was elevated by 38.52, 90.08, 43.16, and 46.78% compared to N0 treatment, respectively. The grain yield of wheat (GY) was significantly influenced by the rates of N fertilizer, and up to the maximum under the N105 treatment (Table 1). The N-fertilizer treatments significantly increased the GY by 21–74%. The NUE ranged from 13.61 to 19.64%, and the N210 treatment had the lowest NUE.

### 3.2. Community Abundances of AOA and AOB

The ammonia-oxidizing archaea (AOA) and bacterial (AOB) abundance were determined by quantifying gene copy number using quantitative PCR (Figure 1). With the increase of N, AOA abundance was decreased in a regular trend from 4.88 × 10^7^ to 1.05 × 10^7^ copies g^−1^ dry soil, while AOB abundance increased from 3.63 × 10^7^ up to a maximum of 8.24 × 10^7^ copies g^−1^ dry soil with the N105 treatment. Compared with the non-N control, the N52.5, N105, N157.5, and N210 treatments significantly increased the AOB community population by 50%, 127%, 116%, and 79%, while decreasing the AOA community population by 48%, 57%, 75%, and 78%, respectively. The ratios of AOA to AOB decreased with increasing N fertilizer rates, ranging from 1.34 to 0.16. 

AOA abundance showed significantly positive correlations with soil pH and NUE, and negative correlation with soil NH_4_-N, NO_3_-N, PNA, and grain yield (Table 2). AOB abundance was positively correlated with soil TN, NO_3_-N, TP, AP, PNA, and grain yield, but negatively correlated with soil pH and NUE. The ratio of AOA/AOB positively correlated with soil pH and NUE, and negatively correlated with soil NH_4_-N, NO_3_-N, AP, PNA, and grain yield. In addition, soil PNA had significant positive correlations with soil TN concentration, NO_3_-N concentration, and grain yield.

### 3.3. Richness and Diversity of AOA and AOB

All AOA sequences with a sequence similarity of 97% were clustered into 48 OTUs, of which 7 OTUs had relative abundance >0.1% (Figure 2). Based on the phylogenetic tree, only 3 AOA OTUs (OTU 1, OTU 2, and OTU 6) could be grouped into one *Candidatus Nitrosocosmicus* (Figure 2a). OTU 1 predominated across all five treatments (up to 97%). Among these 7 OTUs, only the relative abundance of OTU 1 (97.97%) under the N105 treatment increased significantly (Figure 2b).

All AOB with a sequence similarity of 97% were clustered into 693 OTUs, of which 23 OTUs had relative abundance >0.2% (Figure 3). Phylogenetically, only 10 AOB OTUs could be grouped into one *Nitrosospira* cluster 3 (Figure 3a). OTU 1, OTU 6, OTU 8, and OTU 11 were identified as unclassified *Nitrosospira* genus. OTU 2, OTU 4, OTU 9, and OTU 45 were identified as *Nitrosospira briensis* species, while OTU 19 and OTU 22 were identified as *Nitrosospira* species Nsp40. OTU 4 was the most abundant, accounting for 37.9% of the total readings, followed by OTU 9 (13.85%) (Figure 3b). Additionally, among the five treatments, the relative abundance of OTU 12 (*p* = 0.025), OTU 7 (*p* = 0.040), OTU 46 (*p* = 0.009), OTU 45 (*p* = 0.023), and OTU 64 (*p* = 0.013) significantly differed with the N treatments. The relative abundance of OTU 12 was significantly higher under N105 treatment (9.33%) than under the other treatments; the relative abundance of OTU 7 in N210 (1.85%) and N157.5 treatments (2.20%) were significantly lower than that of N0 treatment (5.60%).

N fertilizer rates had significant impacts on the AOB diversity as shown by Chao1, Simpson, and Shannon indexes, rather than AOA diversity (*p* > 0.05) (Figure 4). Among five treatments, the N210 treatment had greater Chao1 for AOA than the N0 treatment (Figure 4a), while the N105 treatment had lower Simpson and Shannon indexes for AOA than the N0 treatment (Figure 4b,c). The N105 and N210 treatments had the greatest and lowest Chao1, Simpson, and Shannon indexes for AOB, respectively (Figure 4d–f).

### 3.4. AOA and AOB Community

As for the AOA community, approximately 100% of AOA reads were assigned to the phylum *Thaumarchaeota*. *Candidatus Nitrosocosmicus* and *Nitrososphaera* were the two dominating AOA genera representing 99.97–99.99% and 0.01–0.02% of the total sequences (Appendix A). Among the five treatments, the composition of the AOA community did not differ significantly. As for the AOB community, 100% of reads were assigned to the phylum *Proteobacteria*. *Nitrosospira* and *Nitrosomonas* were the dominant genera of AOB, representing 99.96–100% and 0.00–0.04% of the total sequences, respectively, while 30.81%–42.90% genus of *Nitrosospira* and 0.00–0.04% genus of *Nitrosomonas* were unclassified (Appendix A). N fertilizer rates had significant impacts on the AOB community. Among the five treatments, N210 and N157.5 showed a higher relative abundance of *Nitrosospira* but a low relative abundance of unclassified *Nitrosospira* compared to those under the N52.5 and no-N control. 

The principal coordinate analysis (PCoA) results showed differences in the community structure of AOA and AOB under different N fertilizer treatments (Appendix A). Total variance for AOA was 34.9% and 13.3% (Appendix A), while variance for AOB was 17.9% and 13.8% (Appendix A), explained by PCoA1 and PCoA2, respectively. There was one major cluster in the community structure of AOB at N105 treatment, while the samples of N0 and N210 treatments had different community composition patterns and were separated from the major cluster (Appendix A). However, no major clusters were found in the community structure of AOA (Appendix A).

### 3.5. Network Associations among OTUs of AOB Species

A correlation-based network was constructed with the OTUs of AOB species to reveal potential co-occurrence between the microbial groups in different N fertilizer treatments (Figure 5). The network showed 52 neighbors, and the average path length was 3.20. The modularity value was only 0.9, suggesting that the AOB network was modular. Among the AOB network, 23 nodes (17% of the total nodes) were assigned to *Nitrosospira briensis* species, 61 nodes (45%) assigned to *Nitrosospira* unclassified species, one node (2%) assigned to *Nitrosospira Np39-19* species (OTU 32), and the other nodes (37%) assigned to null species. However, the OTUs of AOA species failed to form a co-occurrence network based on correlation.

### 3.6. Correlation of Ammonia-Oxidizing Communities with the Soil Properties

The correlation among alpha indices of AOA and AOB diversity, soil properties, and PNA is shown in Table 3. In the AOA community, pH was correlated with Simpson index and negatively correlated with Chao1 and Simpson index; NO_3_-N was positively correlated with chao1 and Observed species index, while NH_4_-N was only positively correlated with Observed species index; both AP and PNA was negatively correlated with Simpson, Shannon, and Pielou_e index. In AOB community, pH was only positively correlated with Pielou_e index; NH_4_-N was negatively correlated with Simpson, Shannon, and Pielou_e index, while NO_3_-N was only negatively correlated with Simpson index; AP was positively correlated with Chao1 and Observed species index; PNA was positively correlated with Chao1 and Shannon index. 

For the AOA community, RDA results showed that the first axis explained 28% variation and the second axis explained 8.2% variation (Figure 6a). Moisture showed a significant influence on the AOA community (72.6%, *p* = 0.020). For the AOB community structure, the first axis explained 31% variation and the second axis explained 7% variation (Figure 6b). The NH_4_-N was the most determinant of structuring the AOB community (67.3%, *p* = 0.004). Thus, AOA and AOB communities were closely related to soil properties.

The correlation between AOA (Figure 7a) and AOB abundance (Figure 7b) at the genera level and soil properties was further determined using a correlation heatmap. Soil NO_3_-N was positively correlated with AOA genus *Nitrososphaera* (*r* = 0.526, *p* = 0.044) and AOB genus *Nitrosospira_sp_Nitrosospira_briensis* (*r* = 0.775, *p* = 0.001), and negatively correlated with AOA genus *Candidatus Nitrosocosmicus* (*r* = −0.526, *p* = 0.044) and AOB genus *Nitrosospira_sp_unclassified* (*r* = −0.777, *p* = 0.001). Soil moisture was positively correlated with AOA genus *Candidatus Nitrosocosmicus* (*r* = 0.587, *p* = 0.021), and negatively correlated with AOA genus *Nitrososphaera* (*r* = −0.587, *p* = 0.021) and AOB genus *Nitrosomonas_sp_unclassified* (*r* = −0.780, *p* = 0.001). Soil NH_4_-N concentration was positively correlated with AOB genus *Nitrosospira_sp_Nitrosospira_briensis* (*r* = 0.773, *p* = 0.001), and negatively correlated with AOB genus *Nitrosospira_sp_unclassified* (*r* = −0.780, *p* = 0.001). Soil pH was positively correlated with AOB genus *Nitrosospira_sp_Nv6x* (*r* = 621, *p* = 0.013) and *Nitrosospira_sp_unclassified* (*r* = 728, *p* = 0.002), and negatively correlated with AOB genus *Nitrosospira_sp_Nitrosospira_briensis* (*r* = −0.731, *p* = 0.002). The soil available phosphorus concentration was only negatively correlated with AOB genus *Nitrosospira_sp_L115* (*r* = −0.599, *p* = 0.018), *Nitrosospira_sp_Np39_19* (*r* = −0.607, *p* = 0.016), and *Nitrosospira_sp_Nl20* (*r* = −0.681, *p* = 0.005). Total nitrogen concentration was negatively correlated with AOB genus *Nitrosospira_sp_PJA1* (*r* = −0.523, *p* = 0.046). In addition, GY was positively correlated with AOB genus *Nitrosospira_sp_Nitrosospira_briensis* (*r* = 0.725, *p* = 0.002), and negatively correlated with AOB genus *Nitrosospira_sp_unclassified* (*r* = −0.709, *p* = 0.003). NUE was positively correlated with AOB genus *Nitrosospira_sp_Nv6x* (*r* = 0.732, *p* = 0.002) and *Nitrosospira_sp_unclassified* (*r* = 0.583, *p* = 0.023), but negatively correlated with AOB genus *Nitrosospira_sp_Nitrosospira_briensis* (*r* = −0.580, *p* = 0.024).

## 4. Discussion

### 4.1. Effect of N Fertilization on AOA and AOB Abundances

Nitrogen (N) addition is crucial for agriculture production and influences the microbial communities [58]. The coexistence of AOA and AOB has been studied in various types of soil [59,60]. This study showed that AOA and AOB communities responded differentially to N fertilizer application rates (Figure 1). The gene copies vary from 1.05 × 10^7^ to 4.88 × 10^7^copies g^−1^ of dry soil for *amoA* AOA and 3.63 × 10^7^ to 8.24 × 10^7^ copies g^−1^ of dry soil for *amoA* AOB. The ratio of AOA to AOB (0.16 to 1.34) was lower than previous studies (17 to 1600) [6,40], which was inconsistent with the previous conclusion that the AOA community dominates agricultural soils [37,38,39]. Compared with the no-N control, the N fertilizer application rate significantly increased AOB abundance (by 50–127%) and decreased AOA abundance (by 48–78%), suggesting that AOB preferred nutrient-rich environments, while AOA was likely more suitable for nutrient-deficient environments [61,62,63]. This study found that N fertilizer rates significantly increased AOB abundance but had little effect on the AOA abundance. 

Inorganic N is used as an energy source for the colonization and growth of the AOB community [18,64]. The significantly positive correlation between AOB abundance and TN further showed that N had an important role in the growth of AOB. Higher ammonia (NH_4_^−^) substrates are beneficial to the growth of AOB, while low ammonia (NH_4_^−^) substrates are beneficial to the growth of AOA [65]. Compared with AOB, AOA was more favorable for lower concentrations of soil NH_4_^+^ and NO_3_^−^, as well as higher pH, which contributed to increased wheat NUE (Table 2). In contrast, increasing NO_3_^−^ concentrations and decreasing soil pH facilitate AOB growth, thereby increasing the yield of wheat grains. Our results contradict the studies of Nicol et al. [66] on grazed grassland soils (pH in the range of 4.9–7.5), which might be because of different soil types. In summary, soil pH, NO_3_-N, and NH_4_-N were the main reasons for the difference in the abundance of AOA and AOB. In addition, soil total phosphorus, total N, and available phosphorus also affected the AOB abundance, while soil moisture did not affect AOA and AOB (Table 2), inconsistent with the study of Yang et al. [67] on irrigated soil.

### 4.2. Composition of AOA and AOB

Previous studies showed a significant impact of N fertilizers on the AOA communities in acidic soil [15,68] and the AOB communities in alkaline sandy soil [37]. The increase in abundance and activity of AOB usually exceeds that of AOA after adding N fertilizer [69,70]. The present study showed that AOA and AOB communities responded differently to N fertilizer application rates. Application of different N rates had significant impacts on the diversity of AOB but no significant effect on the diversity of AOA (Figure 4). In addition, soil samples applied with 105 kg N ha^−1^ had lower diversity of AOA while having greater diversity of AOB (Figure 4), indicating that the optimum N fertilizer rate would increase the diversity of AOB [58]. 

The *Thaumarchaeota* group is widely distributed and the dominant archaea in terrestrial systems [71,72]. We found that approximately 100% of AOA reads belonged to the *Thaumarchaeota*, and 100% of AOB reads belonged to the *Proteobacteria* at the phylum level. At the genera level, *Candidatus Nitrosocosmicus* (>99.9%) was the most dominant AOA genus, and *Nitrosospira* (>99.9%) was the most dominant AOB genus (Appendix A), which was consistent with many previous studies [73,74,75]. Song et al. [76] reported that *Nitrosomonas* generally prefers neutral soils and *Nitrosospira* prefers acidic soils. However, the tested soil was alkaline soils (pH 8.67–8.99), *Nitrosomonas* was also found in AOB but had a low abundance (<0.01%) (Appendix A). *Nitrosospira* is often the most abundant in semi-arid soil [58,77,78]. The study found that at the higher N-treatments (N210 and N157.5), soil had a greater abundance of *Nitrosospira* and lower abundance of unclassified *Nitrosospira* than the lower N-treatment (N52.5) or no-N (N0), suggesting that it is necessary to re-determine the conclusion that *Nitrosospira* prefers low-N environments [79], considering its sub-classification. This study also found a major cluster in the AOB community structure, suggesting that the AOB community is more stable in the soil at the optimum N rate (105 kg N ha^−1^ yr^−1^) (Appendix A). However, AOA communities were relatively insensitive to different rates of N fertilizer, consistent with the study of Tao et al. [39].

Microbial communities interact under different N fertilizer treatments [80]. This result showed that 17% and 45% of total nodes among the keystone species of the AOB network were assigned to *Nitrosospira briensis* and *Nitrosospira* unclassified species, respectively (Figure 5), while AOA species failed to form a co-occurrence network based on correlation. PNA was correlated with the AOB community (with a contribution of 16.6%), but not correlated with the AOA community (Figure 6), indicating that the nitrification in the alkaline soil tested in this study was mainly driven by AOB community. In our study, AOB was the main ammonia-oxidizing community, which was also corroborated in other studies in semi-arid soils [18,46,59].

### 4.3. Relationships between Soil Properties and AOA and AOB Communities

The application of N fertilizers affects many plant-soil characteristics, such as crop yield, nitrogen use efficiency (NUE), and soil properties, which are all important factors that determine the soil microbial habitat [17,81]. Distance-based redundancy analysis (RDA) showed that soil moisture was the most important factor affecting AOA community (72.6% contribution) (Figure 6). Similarly, Yang et al. [67] reported that soil moisture was the dominant factor controlling AOA communities. Moreover, AOA genus *Candidatus Nitrosocosmicus* abundance was also significantly negatively correlated with NO_3_-N concentration (Figure 7), which is also because AOA is more suitable for nutrient-deficient environments [63]. Similarly, soil NH_4_-N was the dominant factor controlling AOB communities (with a contribution of 72%), which showed that N fertilization-induced NH_4_-N concentration changes were the dominant factor controlling the AOB communities. In addition, AOB genus *Nitrosospira* abundance was also positively correlated with NO_3_-N and negatively correlated with soil pH, which is also because AOB prefers nutrient-rich and lower pH environments [15,63,68].

## 5. Conclusions

Long-term different rates of N fertilizer led to the increase of soil NH_4_-N, NO_3_-N concentrations, and AOB abundance, and the decrease of soil pH and AOA abundance. Among soil physicochemical parameters, moisture was the most important contributor to the AOA community, whilst NH_4_-N was the most important contributor to the AOB community. The application of N fertilizer had a more significant impact on the diversity and community of AOB than AOA in this weak alkaline soil. The predominant phyla of AOA and AOB phyla were *Thaumarchaeota* and *Proteobacteria*, respectively. Excessive N fertilization resulted in a significant decrease in the relative abundance of genus *Nitrosospira*. N fertilization rate of 105 kg N ha^−1^ yr^−1^ obtained the greatest AOB diversity, the gene copy of *amoA* AOB (8.24 × 10^7^ copies g^−1^ dry soil), and wheat yield (49% higher than control) for sustainable wheat production in a semi-arid area.

## Figures and Tables

**Figure 1 ijerph-19-02732-f001:**
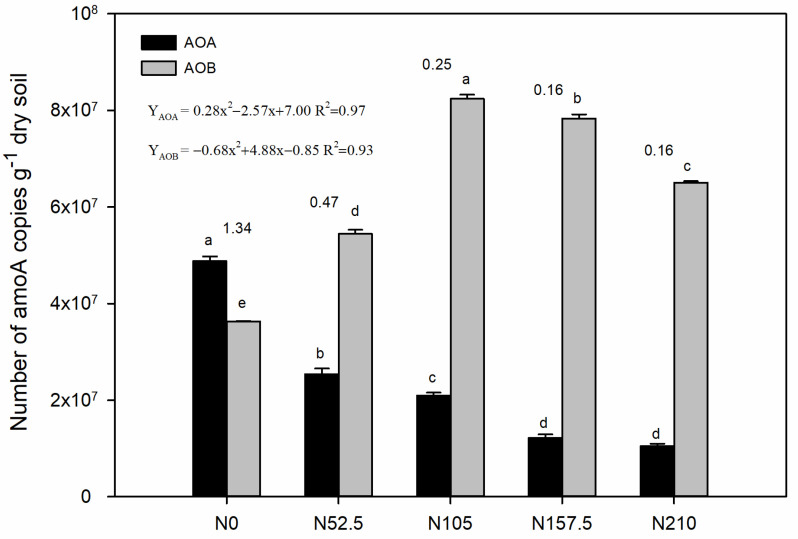
Abundances of ammonium-oxidizing archaea (AOA) and bacteria (AOB). The AOA/AOB ratio of each treatment is shown on the top of the column. N0, non-N-fertilized control; N52.5, N105, N157.5, N210, annual N fertilizer application at 52.5, 105.0, 157.0, and 210.0 kg N ha^−1^, respectively. Error bars indicate the standard errors of the means (*n* = 3). Significant differences are based on AOA and AOB Permanova analyses separately, and the different letters indicate means that are significantly different at *p* < 0.05.

**Figure 2 ijerph-19-02732-f002:**
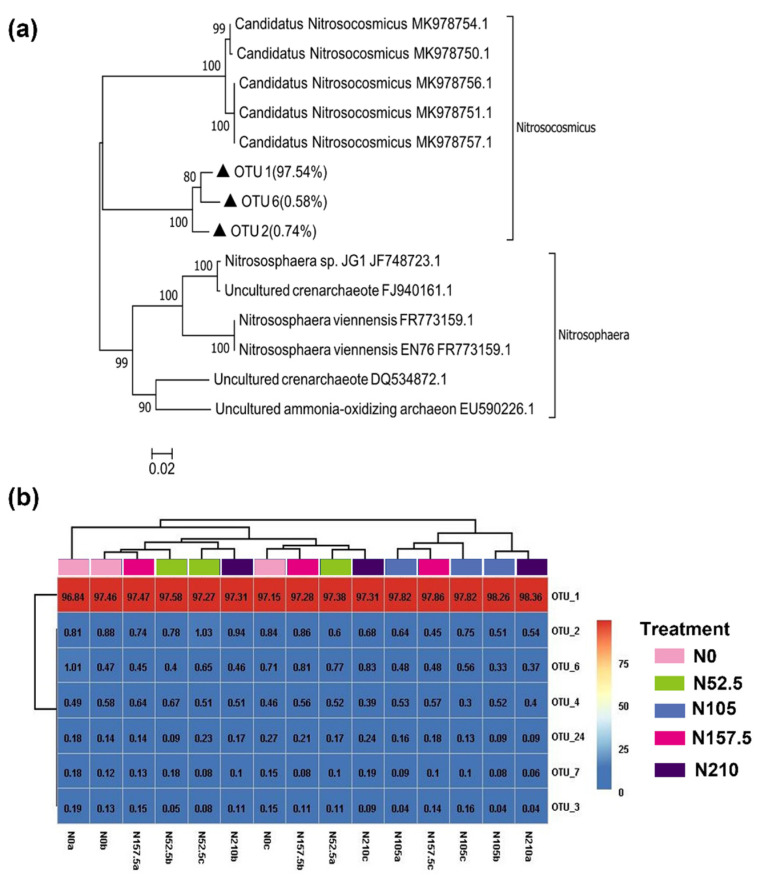
Neighbor-joining phylogenetic tree (**a**) and heatmap (**b**) for representative AOA OTUs (representatives with top 7 relative abundances) under N0, N52.5, N105, N157.5, and N210 treatments. OTUs from this study are depicted by closed triangle. Numbers in brackets represent the total relative abundances of OTUs for each branch. Bootstrap values (>50%) are indicated at branch points. The scale bar represents 2% sequence divergence. The color from blue to red indicates the relative abundance of AOA OTUs from least to most. The number in each text indicates the relative abundance value of each OTU in the sample. a, b, and c represent the first, second, and third experimental replications, respectively, for the corresponding treatment.

**Figure 3 ijerph-19-02732-f003:**
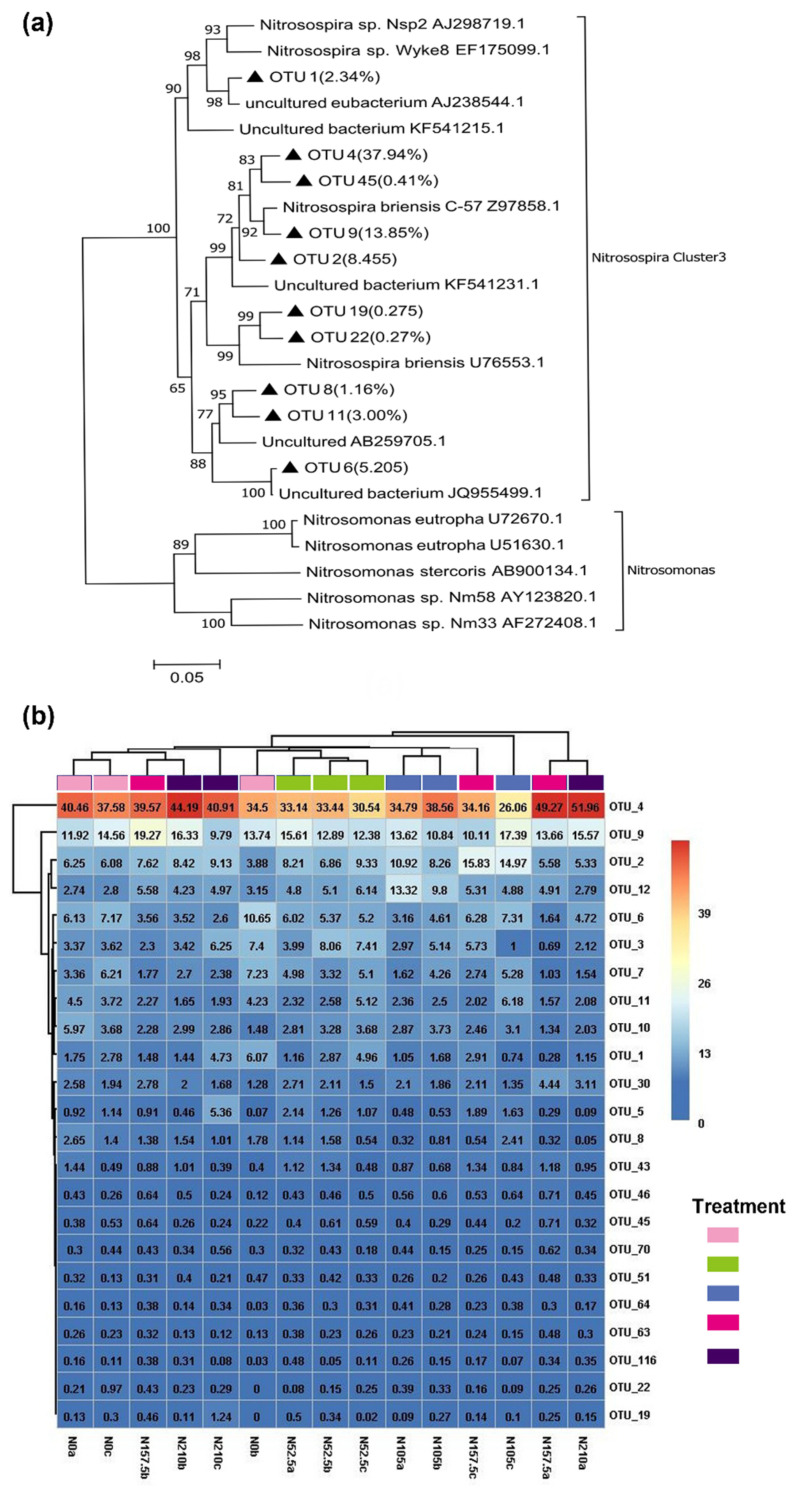
Neighbor-joining phylogenetic tree (**a**) and heatmap (**b**) for representative AOB OTUs (representatives with top 23 relative abundances) under N0, N52.5, N105, N157.5, and N210 treatments. OTUs from this study are depicted by closed triangle. Numbers in brackets represent the total relative abundances of OTUs for each branch. Bootstrap values (>50%) are indicated at branch points. The scale bar represents 5% sequence divergence. The color from blue to red indicates the relative abundance of AOA OTUs from least to most. The number in each text indicates the relative abundance value of each OTU in the sample. a, b, and c represent the first, second, and third experimental replications, respectively, for the corresponding treatment.

**Figure 4 ijerph-19-02732-f004:**
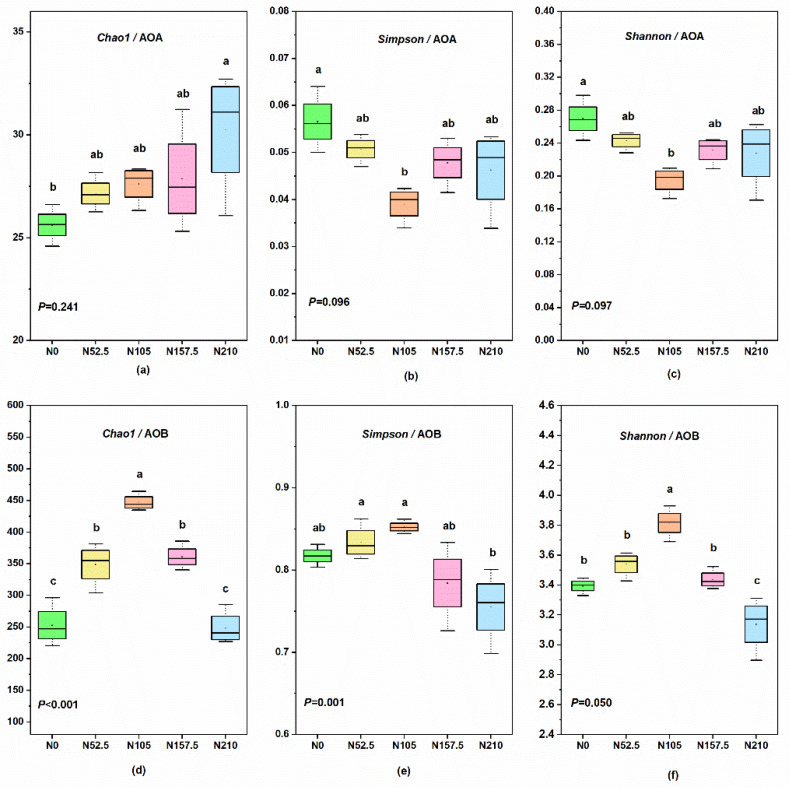
Alpha diversity of AOA and AOB according to the Chao1 (**a**,**d**), Simpson (**b**,**e**), and Shannon (**c**,**f**) indexes as affected by long-term nitrogen (N) fertilizer treatment. N0, non-N-fertilized control; N52.5, N105, N157.5, N210, annual N fertilizer application at 52.5, 105.0, 157.5, and 210.0 kg N ha^−1^, respectively. Boxplots with different letters indicate means that are significantly different at *p* < 0.05.

**Figure 5 ijerph-19-02732-f005:**
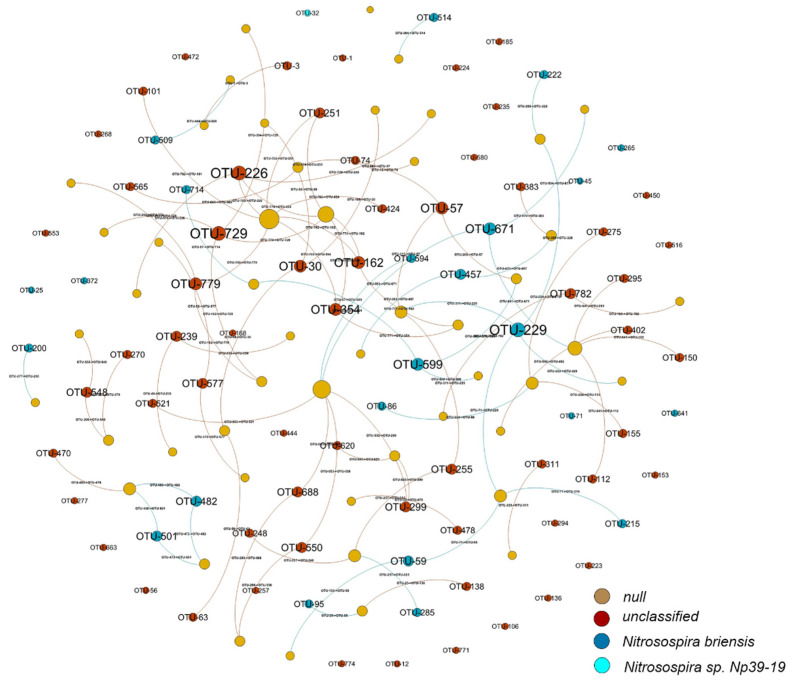
Co-occurrence network based on AOB OTUs across the different N fertilizer treatments. Nodes represent different OTUs, and these OTUs are colored by modules. The size of each node is proportional to the number of connections (its degree). The widths of each connection between two nodes (edges) are scaled according to its weight, and the edge color is derived from the nodes to which they are connected. The label shows the phylogenetic relationship of each node down to the AOB species level.

**Figure 6 ijerph-19-02732-f006:**
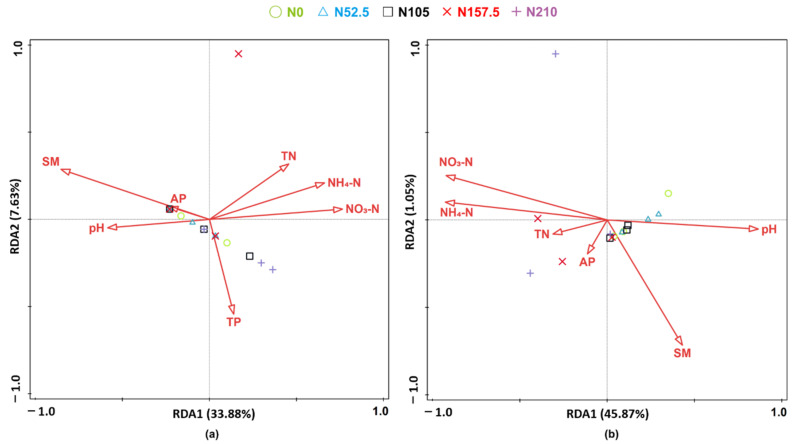
Redundancy analysis (RDA) of (**a**) AOA and (**b**) AOB communities based on soil parameters and PNA. N0, non-N-fertilized control; N52.5, N105, N157.5, N210, annual N fertilizer application at 52.5, 105.0, 157.5, and 210.0 kg N ha^−1^, respectively. pH, soil pH, TN, soil total nitrogen; NH_4_-N, soil ammonium-N; NO_3_-N, soil nitrate-N; TP, soil total phosphorus; AP, soil available phosphorus; SM, soil moisture.

**Figure 7 ijerph-19-02732-f007:**
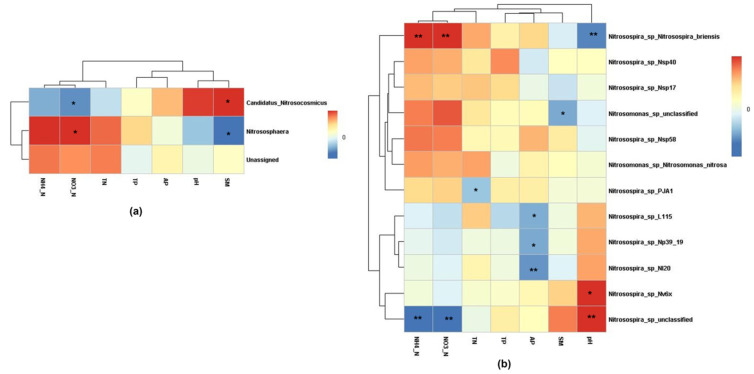
Correlation heatmap of relative abundances of AOA (**a**) and AOB (**b**) and soil properties at the genus level. pH, soil pH, TN, soil total nitrogen; NH_4_-N, soil ammonium-N; NO_3_-N, soil nitrate-N; TP, soil total phosphorus; AP, soil available phosphorus; SM, soil moisture. The shade of color indicates the correlation between relative abundance of each genus and soil parameters. * *p* < 0.05; ** *p* < 0.01.

**Table 1 ijerph-19-02732-t001:** Soil nutrients, potential nitrification activity, wheat grain yield, and nitrogen use efficiency as affected by nitrogen fertilizer treatments.

Treatment	pH	TN (g kg^−1^)	NH_4_-N (mg kg^−1^)	NO_3_-N (mg kg^−1^)	TP (g kg^−1^)	AP (mg kg^−1^)	SM (%)	PNA (mg NO_3_-N kg^−1^ ha^−1^)	GY (kg ha^−1^)	NUE (%)
N0	8.99 ± 0.04a	0.91 ± 0.04a	10.96 ± 0.13b	12.92 ± 0.10e	0.74 ± 0.03a	18.16 ± 1.19b	12.13 ± 0.75a	1.22 ± 0.04c	2696.65 ± 107.12c	-
N52.5	8.81 ± 0.02b	0.95 ± 0.11a	11.06 ± 0.45b	16.23 ± 0.45d	0.82 ± 0.07a	19.99 ± 0.45b	12.07 ± 0.38a	1.69 ± 0.12b	3270.2 ± 144.94b	19.64 ± 2.4a
N105	8.72 ± 0.02bc	1.05 ± 0.02a	11.62 ± 1.32b	21.42 ± 0.52c	0.88 ± 0.05a	22.39 ± 0.17a	11.98 ± 0.22a	2.32 ± 0.15a	4007.88 ± 158.98a	19.07 ± 0.56a
N157.5	8.67 ± 0.01c	1.01 ± 0.03a	16.47 ± 0.21a	30.07 ± 0.56b	0.88 ± 0.07a	19.84 ± 0.57b	11.4 ± 0.41a	1.75 ± 0.12b	4675.52 ± 33.67a	16.3 ± 0.73ab
N210	8.68 ± 0.04c	0.96 ± 0.01a	17.01 ± 0.32a	35.64 ± 0.35a	0.87 ± 0.06a	20.15 ± 0.69b	11.23 ± 0.65a	1.79 ± 0.14b	4209.77 ± 156.71a	13.61 ± 0.68b
ANOVA *p*-value	0.002	0.410	<0.001	<0.001	0.417	0.032	0.647	0.001	<0.001	0.042

N0, non-N-fertilized control; N52.5, N105, N157.5, N210, annual N fertilizer application at 52.5, 105.0, 157.5, and 210.0 kg ha^−1^, respectively; TN, total N; NH_4_-N, ammonium-N; NO_3_-N, nitrate-N; TP, total phosphorus; AP, available phosphorus; SM, soil moisture; PNA, potential nitrification activity; GY, wheat grain yield; NUE, nitrogen use efficiency. Values are means of three replicates ± standard error. Within a column, means followed by different letters are significantly different at *p* < 0.05.

**Table 2 ijerph-19-02732-t002:** Pearson’s correlations analysis between soil properties, grain yield (GY), nitrogen use efficiency (NUE), potential nitrification activity (PNA), and abundances of AOA and AOB.

	pH	TN	NH_4_-N	NO_3_-N	TP	AP	SM	GY	NUE	PNA
AOA	0.900 **	−0.348	−0.728 **	−0.863 **	−0.490	−0.470	0.320	−0.893 **	0.895 **	−0.580 *
AOB	−0.802 **	0.563 *	0.443	0.605 *	0.515 *	0.661 **	−0.225	0.860 **	−0.661 **	0.787 **
AOA/AOB	0.888 **	−0.412	−0.596 *	−0.763 **	−0.513	−0.573 *	0.273	−0.868 **	0.872 **	−0.685 **
PNA	−0.480	0.660 **	0.328	0.674 **	0.320	0.135	−0.015	0.758 **	−0.479	-

The values are correlation coefficients * *p* < 0.05; ** *p* < 0.01.

**Table 3 ijerph-19-02732-t003:** Pearson’s correlation among alpha diversity index of AOA and AOB, soil properties, and potential nitrification activity (PNA).

Item	Alpha Index	pH	TN	NH_4_-N	NO_3_-N	TP	AP	SM	PNA
AOA	Chao1	−0.554 *	0.128	0.450	0.589 *	−0.112	0.284	−0.463	0.080
	Simpson	0.524 *	−0.323	−0.254	−0.330	−0.328	−0.665 **	0.088	−0.700 **
	Shannon	0.510	−0.297	−0.236	−0.309	−0.387	−0.688 **	0.031	−0.713 **
	Pielou_e	0.474	−0.222	−0.262	−0.335	−0.219	−0.639 *	0.122	−0.623 *
	Observed species	−0.545 *	0.157	0.715 **	0.696 **	0.165	0.162	−0.386	0.023
AOB	Chao1	−0.304	0.327	−0.189	−0.095	0.396	0.715 **	0.166	0.703 **
	Simpson	0.412	0.087	−0.711 **	−0.616 *	0.188	0.058	0.144	0.357
	Shannon	0.062	0.422	−0.529 *	−0.448	0.157	0.282	0.111	0.535 *
	Pielou_e	0.530 *	−0.186	−0.578 *	−0.497	0.299	−0.182	0.206	0.241
	Observed species	−0.420	0.618 *	−0.149	−0.095	0.036	0.547 *	0.067	0.488

Correlation coefficients followed by * and ** are significant at *p* < 0.05 and *p* < 0.01, respectively.

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
