# Peer review of "Changes in Ammonia-Oxidizing Archaea and Bacterial Communities and Soil Nitrogen Dynamics in Response to Long-Term Nitrogen Fertilization"

_ijerph, 2022, doi:10.3390/ijerph19052732_

Round 1
Reviewer 1 Report
Regarding the methodology, the authors should specify for soil pH analysis, the type of used eluent and also the soil-eluent ratio.
Also, in results section authors should specify the soil reaction class that has been identified after the measurements and according to their national soil classification.
Reviewer 2 Report
Xu and coworkers present a well written field study on the effect of N fertilization rates on the ammonia-oxidizing microbial community and resulting wheat yield. This study nicely leverages a field experiment in which a controlled range of N fertilizer was applied over a 17-year period. Fertilization rates were then correlated to changes in soil characteristics, ammonia oxidizing bacteria (AOB) and ammonia oxidizing archaea (AOA) abundance/diversity, nitrogen utilization efficiency, and grain yield. The data presented suggest that as fertilization rates increase, soil pH and AOA gene sequences decrease. Conversely, AOB gene sequences increased as fertilization rates increase. These gene abundances suggest that nitrification rates in the alkaline soil analyzed here were driven primarily by AOB instead of AOA. This conclusion is not unexpected based on previous studies indicating that AOA have a stronger affinity for low ammonia concentrations, while AOB have a stronger affinity for high ammonia concentrations. The authors also do not address that amoA gene copy number does not always correlate to ammonia oxidizing activity for AOB. Despite the mostly expected results (the authors do indicate that many studies indicate that AOA dominate in agricultural soils, but these soils are generally not alkaline or heavily fertilized), this study also includes detailed diversity analyses of the AOA and AOB communities detected. The manuscript would also benefit from addressing the following:
Table 1 and Lines 24, 179, 444: The grain yield (GY) is higher for fertilization level N157.5 than N105 (4675 kg/ha versus 4007 kg/ha).
Line 27: I agree that fertilization definitely has a more significant impact on the AOB diversity than AOA diversity, there really is very little diversity at all in the AOA community (see Fig. 2).
Line 59: Please define NUE here. The first time this acronym is described is in a Table legend and then in the Discussion.
Line 61: Many recent studies indicate that some N2O is also released during ammonia oxidation and not just from denitrification.
Lines 71-72: What affect do oxygen levels have on AOA and AOB? Other studies show that AOA survive better under lower O2 concentrations.
Lines 126-127: Reference 49 needs to be moved up to the Arch-amoA26F primer. What about primers that have recently been developed to detect comammox Nitrospira? The AOB primers used here likely miss some of those OTUs.
Lines 128-130: More detail regarding gene copy number determination is needed or a reference to a published technique is required. A plasmid with copies of AOA and AOB genes would be needed to construct a standard curve.
Lines 140-141: I am confused by mentioning that “primers were designed using conserved regions to amplify variable regions of rRNA genes and then sequenced”. The rest of the manuscript suggests that only AOA and AOB genes were targeted.
Line 168: The soil pH for N52.5 actually appears to be higher than that of N157.5 and N210.
Table 1: Columns 3-8 are difficult to read. Extra space between these columns would help discern separate numbers.

Reviewer 3 Report
The paper is of great interest and all the sections are well designed but the conclusions, which are a summary of some results. This final section should be improved reasoning what can be deducted from your data.
English spelling should be carefully revised as some sentences are not well typed. Some examples are:
- Line 53 Replace ...PNA considered... with PNA is considered.
- Line 119. Replace …DNA was using… with …DNA was extracted using…
- Line 376. Do you mean AOB and AOA are favored by lower soil ….?
Apart from that some minor comments:
Introduction.
Lines 49 - 50. Urea is an organic fertilizer
Line 56. Please, explain how N fertilization reduces AOB abundance based on the paper cited.
Lines 73-74. Why N-based fertilization may be a primary driver for AOB?
Lines 79-83. I understand that the hypothesis is that N fertilization affects soil parameters and AOA and AOB abundance. But it is pointed out in the hypothesis that soil parameters affect AOB and AOA abundance.
Material and methods
Line 103. Please, also note the mean annual temperature.
Results
Lines 191. Please, replace …decreased from… with …decreased in a regular trend from…
Line 192. Please replace .. increased from xxx to yyy copies… with …increased from xxx up to a maximum of yyy copies…with the N105 treatment.
Line 195-197. If previously stated is done, then you can delete this now redundant sentence: “ The number ... N105 treatment
Table 1. Define GY and NUE in foornote.
Figure 1. I don’t believe Y axis is in log scale.
In footnote indicate that significant differences are based on AOA and AOB Permanova analyses separately.
Figures 2 & 3. The text are barely readable. Consider separating a and b figures one below the other
Discussion.
Line 396. Please, indicate soil types studied in references 64 and 65.
Lines 409-411. I don’t understand very well where this information came from.
Line 418-419. Sentence: “In summary,… in semiarid soil” is too categorical. However you can affirm that in you study AOB was the main ammonia-oxidizing community, which was also corroborated in other studies in semiarid soils [18,39,52].
Conclusions.
Need to be improved as indicated previously.
Reviewer 4 Report
Refers to the manuscript ijerph-1572418: Changes in ammonia-oxidizing archaea and bacterial communities and soil nitrogen dynamics in response to long-term nitrogen fertilization
In the manuscript, the authors presented the results of long-term very reliable studies on changes in the communities of archaea and ammonia oxidizing bacteria and in the dynamics of nitrogen in the soil in response to nitrogen fertilization.
Detailed comments:
Check and correct grammatical errors and spaces, and format chemical compound names correctly throughout the article.
- Abstract
The Abstract of a good journal article always ends with an outline of the benefits of the research results and recommendations as a solution to the presented problem. Such information is missing in the presented Abstract.
- Keywords
„nitrogen use efficiency”
After these words, please add the abbreviation (NUE), as it appears throughout the article.
- Introduction
In the introduction, I ask that the state of the art review be improved to clearly show the progress beyond the current state of the art. What innovations will be introduced into the literature in this manuscript? What are the gaps in knowledge? The final paragraph of the introductory section always emphasizes the novel aspects of the study with a clear purpose and relevance of the results of the study. It is also recommended to discuss and explain what the appropriate policy should be based on the results of this study.
The introduction should be broadened and redrafted to provide a more comprehensive approach.
- Material and methods
This section has been prepared in detail and is not objectionable. Please correct:
“2.5. Statistica analysis” to 2.5. Statistical analysis.
- Results, discussion and conclusions
The authors analyzed the obtained results very carefully and carefully applied them to the information contained in the literature. Conclusions must be convincing statements about what was found to be innovative, contributing to a strong support of the results and the discussion.
I would recommend accepting this manuscript for publishing after minor corrections.
